# Enhancing Robot Task Planning and Execution through Multi-Layer Large Language Models

**DOI:** 10.3390/s24051687

**Published:** 2024-03-06

**Authors:** Zhirong Luan, Yujun Lai, Rundong Huang, Shuanghao Bai, Yuedi Zhang, Haoran Zhang, Qian Wang

**Affiliations:** 1School of Electrical Engineering, Xi’an University of Technology, Xi’an 710000, China; 2221920082@stu.xaut.edu.cn (Y.L.); 3211712276@stu.xaut.edu.cn (R.H.); wangqian77@xaut.edu.cn (Q.W.); 2College of Artificial Intelligence, Xi’an Jiaotong University, Xi’an 710000, China; baishuanghao@stu.xjtu.edu.cn (S.B.); zyd993@stu.xjtu.edu.cn (Y.Z.); zhr2001@stu.xjtu.edu.cn (H.Z.)

**Keywords:** robots, large language models, natural language, semantic alignment method

## Abstract

Large language models have found utility in the domain of robot task planning and task decomposition. Nevertheless, the direct application of these models for instructing robots in task execution is not without its challenges. Limitations arise in handling more intricate tasks, encountering difficulties in effective interaction with the environment, and facing constraints in the practical executability of machine control instructions directly generated by such models. In response to these challenges, this research advocates for the implementation of a multi-layer large language model to augment a robot’s proficiency in handling complex tasks. The proposed model facilitates a meticulous layer-by-layer decomposition of tasks through the integration of multiple large language models, with the overarching goal of enhancing the accuracy of task planning. Within the task decomposition process, a visual language model is introduced as a sensor for environment perception. The outcomes of this perception process are subsequently assimilated into the large language model, thereby amalgamating the task objectives with environmental information. This integration, in turn, results in the generation of robot motion planning tailored to the specific characteristics of the current environment. Furthermore, to enhance the executability of task planning outputs from the large language model, a semantic alignment method is introduced. This method aligns task planning descriptions with the functional requirements of robot motion, thereby refining the overall compatibility and coherence of the generated instructions. To validate the efficacy of the proposed approach, an experimental platform is established utilizing an intelligent unmanned vehicle. This platform serves as a means to empirically verify the proficiency of the multi-layer large language model in addressing the intricate challenges associated with both robot task planning and execution.

## 1. Introduction

Grounded in experiential learning and knowledge accumulation, humans demonstrate a remarkable ability to comprehend intricate tasks through simple communication. Large language models (LLMs), when subjected to extensive and diverse datasets during training, possess the capability to emulate human-like language understanding and the discernment of human intentions. Exploiting the inherent augmentation capability within large language models enables the decomposition of tasks into multiple subtasks of reduced complexity [1]. This distinctive feature can be harnessed for task planning within robotic systems, ultimately leading to more efficient and seamless human–robot interactions.

The current state of research on robotic task planning, grounded in large language models, remains at its nascent stage. Prevailing studies have predominantly concentrated on tasks characterized by low complexity, such as robotic arm trajectory planning and robotic object handling. While these investigations have contributed significantly to establishing a theoretical framework for large-model-based robot control, they fall short in addressing tasks of elevated complexity. Illustratively, consider the task wherein Bob requests a drink of water from Sam. Sam translates this request into a series of small tasks, encompassing finding a cup, locating a water source, filling the cup, returning to Bob’s location, and passing him the cup. The significance of hydration is often underestimated, and the subdivision of such high-complexity tasks into smaller, manageable components is pivotal. Each subtask should be designed to be straightforward, executed through muscle memory. However, prevailing research has predominantly fixated on smaller and more basic tasks. To navigate the intricacies of more complex tasks [2], the employment of a large language model for robotic task planning on a macro level becomes imperative. Furthermore, the expansion into more intricate tasks necessitates the robot’s ability to adeptly handle and integrate complex environmental information into the task planning process.

In tackling this challenge, our investigation reveals that the direct generation of a robot control code using a large language model (LLM) is impractical, leading to considerable latency and errors. A comprehensive examination of cognitive processes underscores that the precision of outcomes produced by a step-by-step model exceeds that of directly generated results [3]. Consequently, it is advisable for the large language model to transition to a step-by-step mode for optimizing the effectiveness of robot motion planning.

Recognizing the superior aptitude of large language models (LLMs) in understanding semantic-level information and delivering accurate feedback, this paper introduces a multi-layer task decomposition architecture employing large language models. The initial step involves breaking down a complex task into a sequence of low-complexity tasks, resembling a common-sense-like progression, aimed at mitigating the overall complexity and execution difficulty of the task [4]. However, these task sequences remain impractical for direct execution by the robot due to the absence of essential environmental information.

To address this limitation, a visual language model is constructed to sense the physical attributes of environmental information. This model interacts with the information through the large language model, thereby acquiring localized environmental information [5]. Subsequently, a subsequent round of task decomposition ensues, generating fine-grained tasks by amalgamating the acquired environment information with low-complexity tasks. To enable effective robot control, alignment between the decomposed tasks and robot commands is achieved at the semantic level. This alignment ensures that the output of the LLM corresponds seamlessly with the semantic requirements of the robot task commands, achieved through feature vector alignment [6]. Consequently, the large language model can output tasks at the semantic level that precisely control the robot to execute the corresponding actions. The methodology outlined in Figure 1 provides a comprehensive overview of the proposed approach in this paper.

We devised a two-dimensional computable space through the implementation of the “heat map algorithm”. This methodology involves the mapping of visual information onto a 2D computable space, offering insights into the relative positions of objects. The resultant mapping guides the robot’s movements by generating dense, point-like trajectories within the heat map. The real-time generation of this image-level mapping allows for dynamic trajectory adjustments in response to changes in the environment.

It is pertinent to highlight that our approach incorporates the sensing of environmental information through a visual language model, which subsequently feeds this information into a large language model [7]. This collaborative interaction facilitates the generation of environment-specific policies by the large language model, diverging from the reliance on pre-trained policies derived from extensive robot data. As a result, our methodology achieves zero-sample robot control within an open instruction set. The integration of the heat map algorithm into a planning framework, encompassing multiple large language models and visual language models, empowers the robot to comprehend abstract semantic information [8]. This integration not only facilitates accurate task execution but also enables free-form natural language control of robots for tasks of heightened complexity. Concurrently, we substantiate the efficacy of our approach in natural language understanding and the guidance of robot behavior.

Our contributions can be succinctly summarized as follows:

(1) Multi-Layer Task Decomposition: Our architecture uses large language models to guide robot behaviors through natural language, enhancing control in complex tasks.

(2) Integration of Environmental Perception: We employ a visual language model to input environmental information into the large language model, enabling task customization.

(3) Semantic Alignment for Task Control: Using semantic similarity methods, we align natural language descriptions with robot control instructions.

(4) Heat Map Navigation Algorithm: Our novel algorithm generates motion trajectories in a 2D space, guiding realistic robot behaviors.

The remainder of the paper is structured as follows: Section 2 provides an overview of related work; Section 3 delineates the architectural design and methodological principles; and Section 4 expounds upon the experimental methodology and presents an analysis of the results, while Section 5 and Section 6 delve into discussions and summarize the methodology elucidated in this paper.

## 2. Related Work

Natural Language Interaction: Natural languages have been extensively researched for instruction extraction and robot control, where the language is able to make constraints and give behavioral specifications for robot behavior. Tellex et al. [9] described core aspects of language use in robots, including understanding natural language requests, using language to drive learning about the physical world, and engaging in collaborative dialogue with humans. These linguistic specifications can be used to reason about intermediate processes in natural language [10]. Micheli et al. [11] introduced a two-stage process and enhanced the performance of model training by interacting with the environment. Previous work has used classical methods for task sequence extraction, such as lexical analysis and formal logic to disassemble tasks. Thomason et al. [12] designed a mobile robotic dialogue agent that understands human commands through semantic analysis. More often than not, the focus of existing research has shifted from online to offline control of robot motion, with the help of local arithmetic enhancements, capable of executing local end-to-end behavioral patterns [13,14]. Brown et al. [15] trained GPT-3 and demonstrated that large language models can greatly improve the correctness of zero-sample recognition. A great deal of work has revolved around giving robot data the form of building robot datasets through natural language annotation. Model learning as we know it with imitation learning to reinforcement learning, all of these methods require a large amount of data in the form of natural language to generate a model with the robot’s data, and the control of the robot can only be realized by interacting with a large amount of data Jiang et al. [16] argued that the combinatorial nature of language is crucial for learning different sub-skills and systematically generalizing them to new ones. The model of controlling a predictor for behavioral interaction with a robot through linguistic commands is closer to our ideas. Sharma et al. [17] optimized the predictor’s model through supervised learning while generating collision-free trajectories in planar computable space. Huang et al. [18] used the code generation and language interaction capabilities of a pre-trained large language model to introduce the knowledge of the large language model into a three-dimensional computational space to instruct a machine to perform precision actions. In contrast, our work focuses on using the semantic understanding capability of the large language model to align instructions at the semantic level, solving the problem of difficulty in matching the task output from the large language model with the robot control instructions.

Language Models for Robotics: The use of large amounts of robotic data to train language models for solving real-world problems in the physical world is a popular field. A large amount of work has focused on the ability of models to understand natural language and interact with it. Zeng et al. [19] showed that such pre-trained micromodels have generic knowledge and that such models are capable of storing different forms of knowledge from a variety of domains. As a carrier of generic knowledge, the large language model needs to be combined with local scenario information to generate specialized knowledge, and a large amount of work has focused on combining environmental information with the macrolanguage model to enable the model to interact with the environment. Liang et al. [20] demonstrated that the large language model is capable of generating policy code from document strings, and they proposed a hierarchical code generation approach to enable the generation of complex code. Huang et al. [21] implemented robot behaviors by constructing action sequences using the general common sense of the large language model. After obtaining the environment information, the large language model is able to understand the environment information, but still lacks the ability to act, and executing the ability to act requires the large language model to invoke robot motion control commands, which often requires the provision of pre-generated libraries of action commands. Wu et al. [22] found that the large language model has excellent summarization and inductive capabilities and that the large language model summarizes user preferences, generates corresponding motion strategies, and invokes the mechanical base and robotic arm to perform the task of item summarization. In contrast, our focus is on improving the correctness of the large language model in controlling the robot’s behavior so that the robot can understand complex semantic information and perform more complex tasks.

## 3. Method

First, we constructed a multi-layer large language model task decomposition architecture, describing in detail the design details and working principles of the architecture (Section 3.1). We then intervened in the process of fine-grained task decomposition by means of a semantic similarity approach to align the sequence of subtasks of the task decomposition with the atomic tasks we set out to perform (Section 3.2). We then demonstrated the generation of trajectories in 2D space to guide realistic robot behavior by sensing environmental information through visual and linguistic models (Section 3.3).

### 3.1. Multi-Layer Large Language Model Architecture for Task Decomposition

In our architectural framework, we incorporated two large language models, as depicted in Figure 2. The first large language model is responsible for comprehending human instructions and subsequently generating an executable coarse-grained plan for the robot. This process involves task decomposition at the level of common sense. However, it is essential to note that the task sequences produced by the first large language model may not precisely reflect the robot’s behaviors [23]. This discrepancy arises from the inherent limitation of common sense, as it lacks environmental information. Consequently, the task sequences resulting from coarse-grained task decomposition exhibit a deficiency in incorporating the capability for interaction with the physical world environment.

To enhance the adaptability of the generalized knowledge within the large language model across diverse environments, the outcomes of the coarse-grained task decomposition were input into the subsequent functional module. This functional module comprises a large language model and a visual model. The visual model is equipped with an extensive repository of pre-trained a priori knowledge, enabling it to discern the categories of items within an image and thereby acquire information about the items in the environment [24]. It is worth mentioning that both LLMs used in this paper were based on ChatGPT 3.5. Additionally, the VLM, which interacts with the environment, is the OWL-ViT model developed by Google and released as open source. We utilized the APIs of ChatGPT 3.5 and OWL-ViT separately to facilitate information exchange among the various models. The large language model interacts with the visual model to extract environmental information perceived by the latter. Subsequently, it engages in a refined task decomposition process, incorporating the environment information. This iterative decomposition yields more precise fine-grained tasks, aligning with the specific nuances of the environment [25]. The generation of a sequence of fine-grained tasks is achieved by organizing these tasks based on the general knowledge embedded in the large language model.

While these tasks provide a precise linguistic description of the robot’s motion, they encounter a challenge in controlling the robot’s movement due to a misalignment issue between the generated task sequences and the robot’s motion instructions. To facilitate the invocation of robot motion by the large language model, we employed a semantic similarity evaluation method to align task sequences with instruction sequences at the vector level. In this process, the text is first vectorized to represent tasks and instructions in a numerical format [26]. Subsequently, vector normalization is applied to mitigate the influence of text length on the vectors. Finally, at the vector level, tasks and instructions are aligned, ensuring semantic consistency. This alignment process allowed the fine-grained tasks generated by the large language model to be effectively mapped to the robot control instructions, overcoming the challenge of controlling the robot’s motion.

Our approach involved transmitting the video captured by the camera to the visual model for processing, enabling the robot to execute appropriate behaviors upon receiving a command, such as navigating to single or multiple target locations [27]. Based on the recognition results derived from the visual information, a two-dimensional planar heat map is generated. In this heat map, the target location is characterized by a high heat value, while the remaining objects exhibited lower heat values. These heat values diverged toward the periphery, forming a comprehensive heat map. Specifically, in this experiment, the robot is represented in the heat map as solid red and blue blocks. Areas with high heat values are depicted as blocks with a red gradient, while non-target locations are represented by blocks with a blue gradient. Utilizing the principles of a greedy algorithm, we can calculate a trajectory to the target location with the highest heat value. The proposed thermal map is designed for real-time updating, ensuring prompt responsiveness to changes in environmental information [28]. Consequently, the navigation method based on the thermal map can quickly adapt and generate corresponding navigation instructions as the environmental context evolves.

### 3.2. Semantic Similarity-Based Alignment of Task Descriptions with Robot Control Instructions

Controlling robot motion through natural language strategies poses a significant challenge due to potential discrepancies between the task planning output from the large language model and the corresponding motion control functions of the robot. The content of the task planning output by the large language model exhibits variations in specific descriptions, introducing ambiguity in the understanding of the specific actions the robot needs to perform. To address this challenge, we employed semantic similarity, emphasizing semantic alignment rather than textual similarity [29], to correlate the task planning with the robot motion control instructions. Recognizing that some fine-grained tasks may require further decomposition to reach a machine-executable level, we proposed a cyclic semantic alignment method. This method aims to iteratively refine the alignment process, enhancing the correspondence between the nuanced task descriptions and the robot’s motion control instructions.

In executable task planning, the task description output from the large language model and the robot control instruction are essentially two semantically similar texts, and we usually used cosine similarity to judge when measuring the similarity of the two texts. The semantic similarity can be computed by first embedding the text in the feature space, and then performing the similarity computation in the feature space [30]. Specifically, this paper used cosine similarity to compute semantic similarity.

We vectorized the text, assuming that the task output from the large language model is text A and the robot control command is text B. We first represented them as vectors $v_{A}$ and $v_{B}$, which are vectorized using the word embedding method, and these vectors represent the position of the text in the vector space.
(1)$$\begin{array}{rr} v_{A} & =\left(w_{A1},w_{A2},\ldots ,w_{An}\right)\\ v_{B} & =\left(w_{B1},w_{B2},\ldots ,w_{Bm}\right) \end{array}$$
where n and m denote the size of the vocabulary in texts A and B, respectively, and $w_{Ai}$ and $w_{Bi}$ are the corresponding vocabulary weights.

In order to remove the effect of text length, the text vector can be normalized. The normalized vectors are denoted as $u_{A}$ and $u_{B}$:(2)$$\begin{array}{cc} u_{A} & =\frac{v_{A}}{∥v_{A}∥}\\ u_{B} & =\frac{v_{B}}{∥v_{B}∥} \end{array}$$

In Equation (2), $u_{A}$ and $u_{B}$ represent the normalized text vectors, while $∥v_{A}∥$ and $∥v_{B}∥$ represent the norms of vectors $v_{A}$ and $v_{B}$, respectively. The normalization operation essentially involves dividing each element in the vector by the vector’s norm.

The effect of doing this is that, regardless of the original length of the text vector, the normalized text vectors all have unit lengths. As a result, when computing distances or comparing similarities between text vectors, they are not affected by the length of the text, thus allowing for better comparison of text similarities.

Cosine similarity is measured by calculating the dot product of two vectors and dividing by the product of the norms of the two vectors. The cosine similarity formula is as follows:(3)$$\mathrm{Similarity}\left(u_{A},u_{B}\right)=\frac{u_{A}\cdot u_{B}}{∥u_{A}∥\cdot ∥u_{B}∥}$$
(4)$$\mathrm{Similarity}\left(u_{A},u_{B}\right)=\frac{\sum_{i=1}^{n}w_{Ai}\cdot w_{Bi}}{\sqrt{\sum_{i=1}^{n}w_{Ai}^{2}}\cdot \sqrt{\sum_{i=1}^{m}w_{Bi}^{2}}}$$

With text similarity matching, we were able to perform alignment between tasks and instructions. The detailed procedure is in Algorithm 1.

In the process of generating task plans, one situation that may occur is that the tasks generated by the large language model in conjunction with the environmental information are incorrect, and this error results in the semantics of the tasks not being able to be aligned with the semantics of the commands, leading to the inability to invoke the robot control commands [31]. For text that cannot be aligned at the semantic level, we fed the task output from the large language model back into the model for a new round of decomposition to re-generate the task sequence, and this process continued until the latest coarse-grained task is fed into the model [32].
**Algorithm 1: Semantic Information Vector Space Alignment Methods****Cosine similarity vector alignment (outline)****1. Input:** text message**2. quantitative:****3.**     vA=wA1,wA2,…,wAnvB=wB1,wB2,…,wBm //vectorization and location information**4. normalization:****5.**     uA=vA∥vA∥uB=vB∥vB∥ //vector normalization**6. cosine similarity:****7.**     SimilarityuA,uB=uA·uB∥uA∥·∥uB∥ //Text Similarity Determination**8.**     SimilarityuA,uB=∑i=1nwAi·wBi∑i=1nwAi2·∑i=1mwBi2 //Alignment of text A and text B**9. Determining text similarity:** (−1 to 1) The closer to 1 the vectors are, the more similar they are.

### 3.3. Robot Heatmap Navigation Algorithm for Open Environment Awareness

Based on Section 3.1 and Section 3.2, we modeled the behavior of a robot controlled by an open natural language L, e.g., by having the robot go first to location A and then to location B. However, generating a robot trajectory based on L is very difficult because the information in L is too granular and lacks the detailed process of the task and, at the same time, lacks information about the environment [32]. Considering that the environment in which the robot works is not static, it is necessary to allow the robot to consider real-time environmental information when performing tasks. Assuming that the large language model decomposes L into several subtasks $\left(l_{1},l_{2},\ldots ,l_{n}\right)$, at this stage of task generation, we concentrated on the real-time environment to create detailed tasks $l_{i}$ to generate motion control commands that the robot can execute [33]. The above approach decomposes complex tasks into subtasks of lower complexity and senses the environment for each of the low-complexity tasks. The core problem of this subsection is how to make full use of the environment information to generate the motion trajectory $\tau_{i}^{r}$ for the robot for each task $l_{i}$. In this paper, the real robots that execute the motion trajectory $\tau_{i}^{r}$ are multiple McNamee-wheeled unmanned vehicles, and with reference to the work of voxposer [18], we combined the environment information with the trajectory generation problem and summarized the problem as follows:(5)$${min}_{\tau_{i}^{r}}\{F_{task}\left(T_{i},l_{i}\right)+F_{control}\left(\tau_{i}^{r}\right)\}\;\;\;\;\mathrm{subject}\;\mathrm{to}\;\;\;\;C\left(T_{i}\right)$$
where $T_{i}$ is an environmental sensing information, $\tau_{i}^{r}\subseteq T_{i}$ is the trajectory of the unmanned cart in the dynamic environment, $C\left(T_{i}\right)$ denotes the constraints of the unmanned cart in the dynamic environment, $F_{task}\left(T_{i},l_{i}\right)$ denotes the completion of the corresponding task within the confines of the dynamic environment, and $F_{control}\left(\tau_{i}^{r}\right)$ specifies the control cost desired by the shortest path or the least task execution time.

It is very difficult to compute $F_{task}\left(T_{i},l_{i}\right)$ based on open natural language, on the one hand because of the problem of difficult alignment between the natural language and the robot task $l_{i}$, and on the other hand, because of the lack of dynamic environment information and real-time robot position. In this regard, we provided a 2D computable space describing the relative position information of objects in a dynamic environment $V\in R^{w\times h}$. We called it a calorific heatmap. It reflects objects in the environment that we are interested in or not interested in [34]; for objects of interest, we defined a high heat value for them, and objects that are not of interest are reflected in the heat map as low heat values. It directs the movement of objects with high heat values in the environment, creating trajectory curves between the robot and the objects of interest. The heat map assigns heat values to various objects in the surroundings. Using sub-tasks defined by the large language model, the task objective is labeled with a high heat value, attracting the robot toward the target area. Objects not of interest have low heat values on the heat map, repelling the robot and guiding it away from non-target areas.

We denoted the high calorific heat target as e and the robot trajectory as $\tau^{e}$. For the subtask $l_{i}$ in $F_{task}\left(T_{i},l_{i}\right)$, we can numericalize the task in the two-dimensional space $V\in R^{w\times h}$ by means of a calorific heatmap. The corresponding task $F_{task}$ in the environment can be approximated by the continuous accumulation of e in the two-dimensional space $V\in R^{w\times h}$. The formula is as follows:(6)$$F_{task}=-\sum_{j=1}^{\left|\tau_{i}^{e}\right|}V\left(p_{j}^{e}\right),\;where\;p_{j}^{e}\in N^{2}\;\mathrm{is}\;e\;\mathrm{discrete}\;\mathrm{position}\;(x,y)\;\mathrm{in}\;\mathrm{step}\;j.$$

Large language models (LLMs) exhibit the capability to adapt their output in response to contextual information. We can influence the LLM to generate content aligning with our preferences through a prompting approach. In this study, prompt engineering is implemented through question-and-answer pairs, comprising questions and results, along with related objects and corresponding questions [35]. Illustrated in Figure 3, for the robot’s shortest path-planning problem, we integrated information about objects in the environment. Using a second LLM, we decomposed the problem into a fine-grained task sequence. The resulting task sequences incorporated environmental information, empowering the robot to execute tasks in a real-world setting.

The prompt-engineered large language model is capable of recognizing objects of interest and understanding the relative spatial information in order to generate motion strategies [36]. Specifically, it can (1) perceive the environmental information by calling the visual model Application Programming Interface (API) to obtain the relative position information of the scene objects; (2) generate the task for the specialized scene based on the perceived environmental information combined with the generic knowledge of the first large language model; and (3) input the tasks into the heat map module through code form to generate the robot motion trajectory $\tau^{e}$ corresponding to each step of the task $l_{i}$. The heat map $V_{i}^{t}=heatmap\left(o^{t},l_{i}\right)$ can be further obtained, where $o^{t}$ is the camera observation at the moment *t* and $l_{i}$ is the task being executed.

In order to be able to generate a smooth trajectory for the robot to travel to the target area, we represented each step of the task as a mathematical problem FtaskTi,li. The motion trajectory can now be planned through the problem defined in Equation (1). The heat map reveals item properties and their positions in the scene. Objects of interest have high heat values, drawing the robot toward them, while non-target items are seen as obstacles with low heat values, pushing the robot away. All positions in the heat map have computable heat values. Our goal is to create a trajectory with the highest heat value, capable of reaching specified subtask locations. We defined the path’s “heat value” as the sum of heat values of all nodes on the path.
(7)RP=∑i=1kHvi
where k denotes the path length and Hv denotes the heat value of node v. We wished to find a path P that maximises the value of RP. A greedy idea was used to select the next node vi in each step such that Hvi is maximal, which can be expressed as:(8)vi=argmaxv∈Nvi−1Hv
where Nvi−1 denotes the set of neighbouring nodes of node vi−1. In this way, we can find a path P={v1,v2,…,vk}, where v1 is the starting point, and vi chosen at each step makes Hvi maximal until the node with the highest calorific value is reached. We provided dense rewards in the space to generate a planning path. During robot operation, because 2D visual information does not contain complete spatial information, it provides a positional relationship between the robot and the environment relative to each other. The robot continuously approximates our generated motion trajectory through this real-time feedback of the heat map [37]. Specifically, the process of updating the environment information in real time provides continuous feedback to control the robot motion nodes so that the robot’s motion profile continuously approaches our generated path trajectory, and when the robot’s behavior is shifted, it is possible to replan the real robot motion trajectory through this feedback. Please refer to Algorithm 2 for more specific details about the algorithm.
**Algorithm 2: Dynamic navigation algorithms for calorific heat maps****Environmental Interaction and Mathematical Representation (outline)****1. Input:** natural language representation task L**2. Breakdown of tasks:****3.**     l1,l2,…,ln**4. The problem is reduced to the optimization equation****5.** minτir{FtaskTi,li+Fcontrolτir}   subject to   CTi**6. Constructing a mathematical representation of** Ftask**7.**     Ftask=−∑j=1τieVpje, where pje∈N2**8. Construct a two-dimensional space:** V∈Rw×h**9. Constructing calorific heat maps:****10.**    Vit=heatmapot,li**11. Define a high calorific value path:****12.**     RP=∑i=1kHvi**//**We define the heat value of a path as the sum of the heat values of all nodes on the path.**13. Path generation:****14.**    vi=argmaxv∈Nvi−1Hv**//**Nvi−1 denotes the set of neighbouring nodes of node vi−1**15. Output:** P={v1,v2,…,vk}             //Path with the highest calorific value

## 4. Experiments and Analyses

We first discuss the design and execution of the holistic experiment and implement the holistic experiment in a real-world environment (Section 4.1). In order to make the holistic task planning more interpretable, we added an intermediate process of task planning, which allows humans to intervene in the intermediate process if the results do not meet human expectations (Section 4.2). We conducted semantic similarity-based feedback experiments and discussed the optimal number of loops (Section 4.3). We presented some failed cases from the experiment and conducted discussions and analyses on these cases (Section 4.4).

### 4.1. Experimental Design

In this experiment, we constructed a real-world experimental platform to support the theory proposed in this paper, and the large language model and the visual model were deployed in the server to extract the generic knowledge by calling API [38]. In this experiment, we used three unmanned carts and a drone to achieve the motion control of the robots through the Ros system. Specifically, the drone provides visual information; the visual information is transmitted back to the server to perceive the environment information through the visual model; and the large language model combines the environment information with the generic knowledge to generate the robot motion scheme to cope with the environment [39] and then generates the optimized trajectory through the heat map algorithm to guide the robots to complete the corresponding tasks as shown in Figure 4.

In Figure 5, we present a detailed design of the experiment, illustrating the intermediate process from the task given in natural language to the generation of heat maps depicting the movement trajectories of robotic; As depicted in Figure 5, the video is captured by the camera mounted on the UAV, providing a global view. Extracting environment information by intercepting images from the video allows us to discern details about the items in the surroundings. The large-scale language model receives a natural language task from a human and, in conjunction with the information extracted by the visual model, decomposes the task and collaboratively generates a sequence of subtasks [40]. This process breaks down the complex task into steps executable by the robot and performs semantic similarity matching in vector space to align with the robot instructions. Trajectories for each task step are generated in a heat map, enabling the robot to approach the target using relative position information provided by these trajectories. Continuous correction of offset through visual information feedback ensures successful execution of the navigation task. For clarity, we used color blocks to cover the unmanned vehicle throughout the experiment, facilitating observation. In the task illustrated in Figure 5, the objective is for the vehicle to initially reach the area where the yellow color block is located and subsequently reach the area where the green color block is situated. To enhance visual representation of the experimental flow, the vehicle is covered with red color blocks in this particular task.

Figure 6 displays five tasks of varying difficulty levels designed by us. We described these five tasks using natural language and employed the framework proposed in this paper to enable the robot to accomplish navigation tasks. It is worth mentioning that all five tasks are types of robot navigation tasks. We progressively increased the complexity of navigation tasks from simple to complex. These tasks required the LLMs to understand natural language and make accurate judgments. In the navigation tasks, we introduced obstacle detection conditions and multi-robot multi-target navigation, covering various difficulty levels of the experimental scenarios. We (1) had the robot arrive at two specified target locations in succession; (2) had the robot arrive at three specified target locations in succession; (3) planned the shortest path through the human-specified locations, focusing on judging whether the strategy generated by the large language model is consistent with the robot’s execution; (4) determined whether there is a human-specified marker in the field and travelled to a specified location A if there is, and to a specified location B if there is not; and (5) designed a task rich in complex semantic information that instructs three unmanned vehicles to perform multi-robot, multi-objective navigation tasks that incorporate and temporally sequence task judgment conditions. Specifically, unmanned vehicle A and unmanned vehicle B each travel to a different human-specified location, and unmanned vehicle C judges that vehicle A has arrived at the specified location before starting to travel to a new location. It is worth noting that all of the above tasks can be interacted with real robots using natural language, and dynamic feedback can be provided in real time to reduce interference caused by changes in environmental information. We successfully completed the above five experiments in a real environment, and the images in Figure 6 were all captured during the actual experimental process. The experiments show that the task decomposition of the large language model, with the addition of environmental awareness and semantic alignment, can control the robots to perform tasks of higher complexity.

### 4.2. Intermediate Process of Task Planning

We used Figure 7 to visually illustrate the output formats of the two LLMs. This enables controllability of each module’s output, allowing for manual intervention in a specific module to generate results that align with our expectations. Introducing an intermediate step of task planning can enhance the interpretability of the overall task planning process. Simultaneously, this intermediate step opens up the opportunity for human intervention in the task planning process, particularly when the generated results deviate from the intended human task execution process [40]. Human intervention is facilitated across three dimensions: coarse-grained task decomposition, fine-grained task decomposition, and motion control instructions [41]. Foreseeably, by controlling the robot through a human-in-the-loop model built upon the foundation laid in this paper, we can enhance the reliability and safety of the robot’s control.

Figure 7 illustrates an example of the intermediate task planning process, wherein the given problem involves selecting a robot to traverse all yellow objects. The task is decomposed, and a detailed intermediate process is displayed. Initially, the first large language model (LLM) conducts a coarse-grained decomposition of the problem, generating a sequence of tasks devoid of environmental information. These tasks involve finding the yellow objects and locating the robot to execute the tasks, followed by sequential movements of the robots to the positions of the yellow objects. In this process, lacking environmental information, these decomposed tasks are answered based on the information contained in the question. The second LLM utilizes environmental information and incorporates the fine-grained task breakdown from sequences produced by the first LLM. This integration results in new task sequences enriched with scene information. It is evident that the LLM can generate more realistic task sequences by incorporating environmental information. Subsequently, by aligning with corresponding motion control commands, a motion trajectory is generated in the heat map to guide the robot in executing the task.

### 4.3. Feedback Experiments Based on Semantic Similarity

The output of semantically similar results by the large language model indicates its ability to comprehend the problem and generate relatively accurate results, which are already acceptable at the semantic level [42]. We facilitated the mapping of the results generated by the large language model to robot control commands through vector alignment. It is noteworthy that when alignment is conducted at the semantic level, the large language model is proficient in decomposing semantically correct tasks [43]. Consequently, these tasks can be successfully mapped to corresponding motion commands through semantic-level alignment.

We conducted a feedback experiment based on semantic similarity and explored the optimal number of iterations. Throughout the experimentation process, we observed that the results generated by the large language model are not consistently optimal. This inconsistency stems from the fact that the process of prompting the large language model does not encompass all working conditions [44]. Consequently, when describing certain complex tasks with detailed natural language, the large language model may face challenges in understanding, leading to difficulties in comprehending intricate task nuances [45]. As a result of the understanding deviation in the problem formulation process, the output results of the large language model may occasionally deviate from our anticipated direction [46].

As shown in Figure 8, we provide feedback on the tasks that posed challenges in aligning at the semantic level. Subsequently, we reintroduced the erroneous task sequences into the second macro-language model for a new round of task decomposition. This iterative process serves to semantically bias the macro-language model toward our cue words by incorporating feedback information about the erroneous tasks [47]. Given that the cue words encompass robot motion instructions, this bias encourages the macro-language model to generate natural language that aligns with the correct instructions enriched with semantic information. This approach enhances the likelihood of outputting correct command results at the semantic level.

To illustrate this process in detail, as shown in Figure 9, we present an example of prompts for the second LLM. This segment indicates that when the LLM receives feedback indicating semantic mismatch, the next output will prefer the format of prompt words we designed. This pattern is advantageous for the LLM to generate outputs biased toward our desired expectations.

As indicated in Table 1, the feedback proves advantageous in improving the success rate of task execution. By informing the large language model about the inaccuracies in task decompositions through feedback, it effectively reduces information uncertainty and steers the decomposition preference toward our cue words [48,49]. The experimental results reveal that the correctness rate reaches 78% after the third feedback. From the fourth feedback onwards, the increase in task success becomes slow. We considered three or four iterations as the optimal number of feedback attempts, demonstrating high system efficiency.

### 4.4. Discussion and Analysis of Failed Cases

During the experiment, there may be reasons leading to the failure of the experiment, specifically when the robot fails to execute the correct tasks. We analyzed these failed cases, which resulted from either the LLMs misinterpreting natural language or the VLM detecting targets incorrectly during the environmental perception process. Because the motion command is aimed at directing the robot to a specific location, which is relatively simple natural language, errors in semantic similarity modules often occur due to incorrect semantic information provided by the LLMs and VLM or incorrect target detections. Therefore, we focused on discussing the failures caused by these two modules, the LLMs and VLM.

When transmitting natural language to the LLMs, it is essential for the LLMs to correctly understand the semantics of the natural language and generate the correct strategy. However, the LLMs cannot always produce accurate results. As shown in Figure 10, when given a task like “Select a robot to pass through all yellow-colored blocks,” the correct task decomposition logic should be for the robot to pass through blocks it has not reached yet. However, the LLMs occasionally generate confused task decomposition processes, such as generating a task to go to a yellow-colored block, which is obviously incorrect. Such logical errors in the task decomposition process can result in the robot failing to execute the task successfully.

Another scenario is when the VLM makes errors during environmental perception, as shown in Figure 11. The VLM may misidentify targets in the presence of changes in light, leading the LLMs to receive incorrect environmental information, resulting in the failure of the robot to execute tasks. Alternatively, the VLM may identify irrelevant items in the scene that users do not want it to recognize, causing interference in the experimental process and potentially generating incorrect results in the heat map.

## 5. Discussion

In this work, we achieved semantic-level alignment between the output of the large language model and the robot control commands and experimentally validated that the approach is capable of controlling a robot using natural language for tasks of higher complexity [50]. The method proposed in this paper enables easier human–robot interaction to accomplish corresponding tasks in the real world, without requiring extensive expertise or skills. Humans can describe task requirements in natural language, and using the framework designed in this paper, the tasks can be understood at the semantic level and translated into task sequences that the robot can execute, thereby driving the robot to achieve task goals. Importantly, this method, through the form of multi-layer task decomposition and interaction with the environment, enables robots to understand and complete tasks with complex semantic information. Compared to traditional natural language-controlled robot technologies that can only handle relatively simple tasks, this method enables robots to perform more complex and abstract tasks.

This work has some limitations. First, this experiment relies on the environment perception module to obtain environment information, which will be limited by the perception module when the visual language model analyzes the object properties. Second, this work performs a numerical analysis in a two-dimensional computable space, which does not provide the robot with high-precision environmental information and constrains the robot from performing more detailed tasks [51]. The third point is that this work designs a feedback mechanism during the task decomposition process, although it solves the problem of aligning the task decomposition with the robot control instructions. But it does not guarantee that the results generated by the large language model do not have logic problems [52]. The robot displacement caused by such logic errors can only be adjusted by the real-time feedback of visual information, and the system will make the robot re-plan the correct trajectory through subsequent task sequences. This process increases the time for task execution.

We can enhance the specificity of the model output through fine-tuning. Specifically, we can use the self-instruct method [53] to generate corpora for each model in the multi-layered large model architecture. We can then fine-tune the respective models using these corpora to improve the accuracy of task decomposition.

We can utilize additional environmental perception modules, which will help to better transform the open environment into a computable space for guiding precise robot actions. Because we are currently only using cameras, in the future, we can enhance the complexity of numerical space by adding different sensors, such as LiDAR and depth cameras, and fusing multimodal sensor information.

In future work, firstly, we can add the multi-environmental sensing module, which can obtain more three-dimensional and rich environmental information through multi-modal environmental sensing. Secondly, we can construct more complex numerical spaces to optimize the motion control strategy so that the robot can perform more delicate tasks. Finally, we will also design prompt words with more semantic information and use prompt engineering to reduce the number of cycles in the task decomposition process and optimize the model system.

## 6. Conclusions

In this research, we present a novel methodology that integrates a large language model (LLM) with a visual language model (VLM) and a calorific heat map. This approach facilitated a multi-layer decomposition of tasks, thereby elevating the precision of task decomposition. Additionally, we introduced an intermediate task planning process to bolster the reliability of robot control. To ensure alignment between LLM outputs and motion control instructions, a vector alignment method is employed. Through rigorous testing and evaluation in a real-world robot scenario, our findings substantiate that the proposed methodology enhances the LLM’s proficiency in comprehending intricate real-world tasks. Furthermore, it amplifies the likelihood of aligning the LLM outputs with motion control commands.

## Figures and Tables

**Figure 1 sensors-24-01687-f001:**
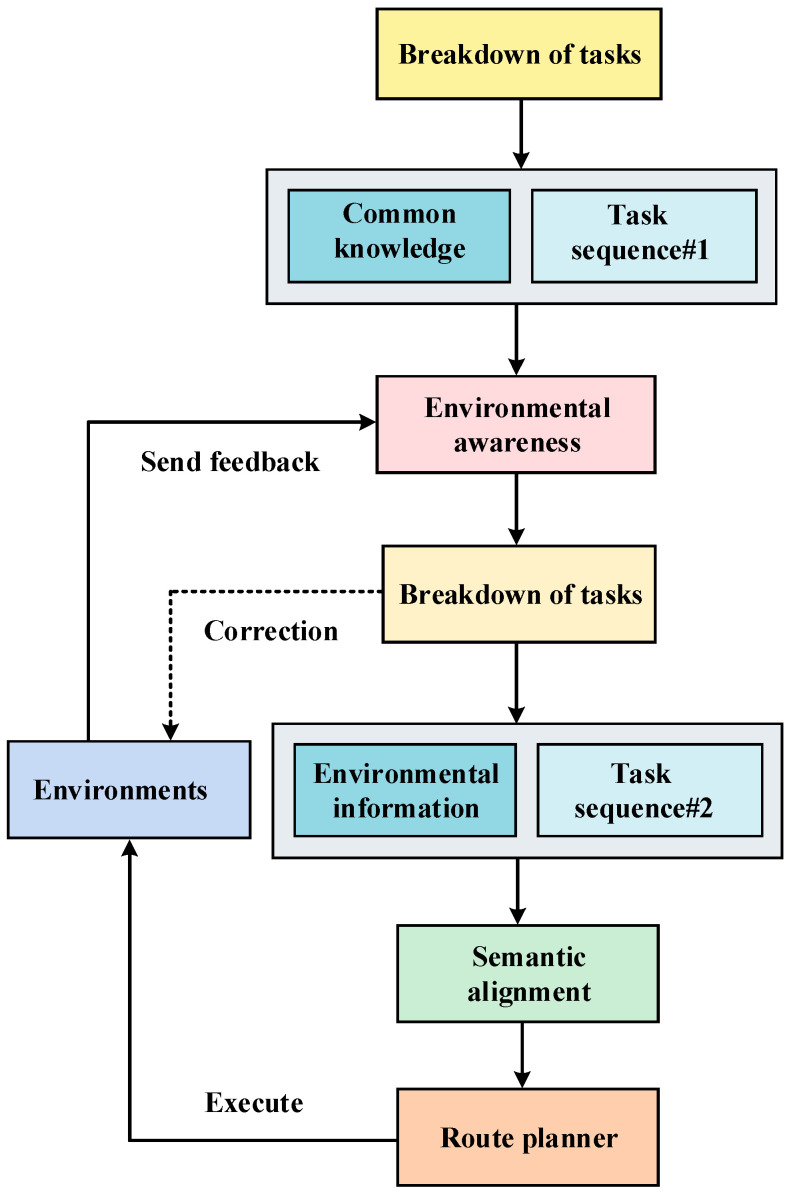
Overview of our approach.

**Figure 2 sensors-24-01687-f002:**
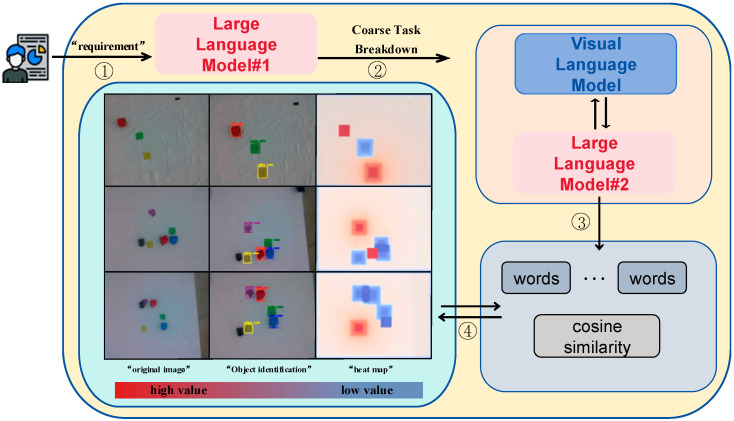
Overall structure and various functional modules.

**Figure 3 sensors-24-01687-f003:**
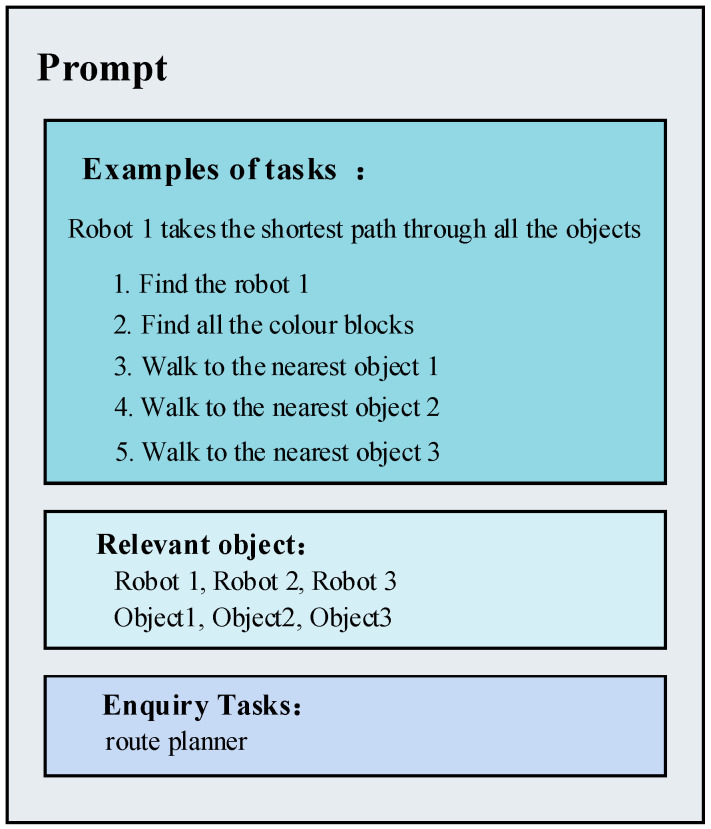
Example of a prompt engineering.

**Figure 4 sensors-24-01687-f004:**
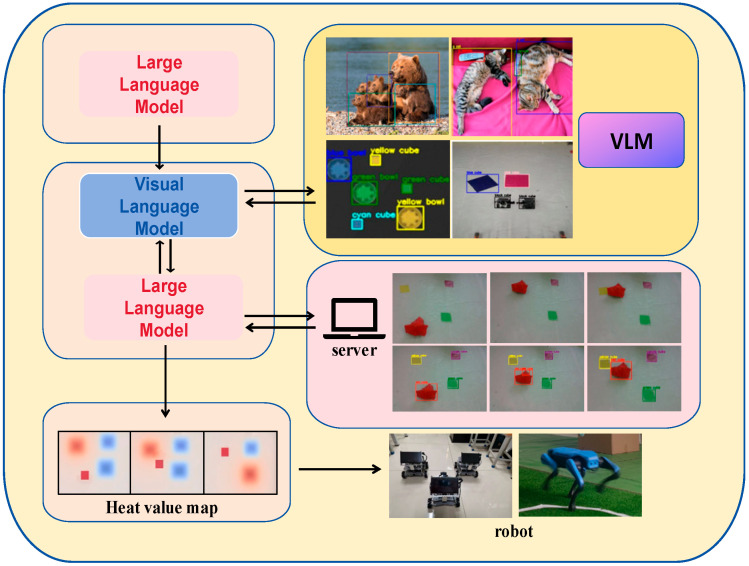
Overall structure of the experiment. In the figure we have used VLM to recognize real images and determined three different categories of target objects, robots, and obstacles, and mapped these categories of objects differently so that they are represented differently in the heat map.

**Figure 5 sensors-24-01687-f005:**
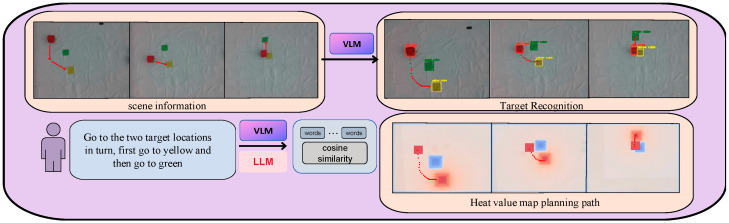
Experimental design that describes the process of generating calorimetric map motion trajectories from image information.

**Figure 6 sensors-24-01687-f006:**
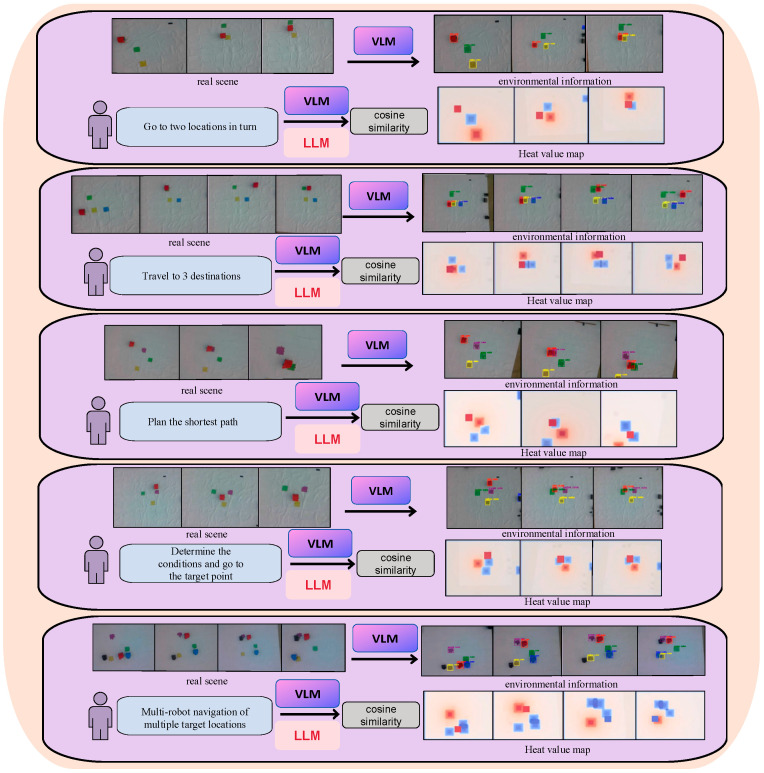
The 5 experiments are designed to include tasks of varying complexity. The flow of the 5 experiments is described, including our process of target recognition from raw image information to finally generating a heat map.

**Figure 7 sensors-24-01687-f007:**
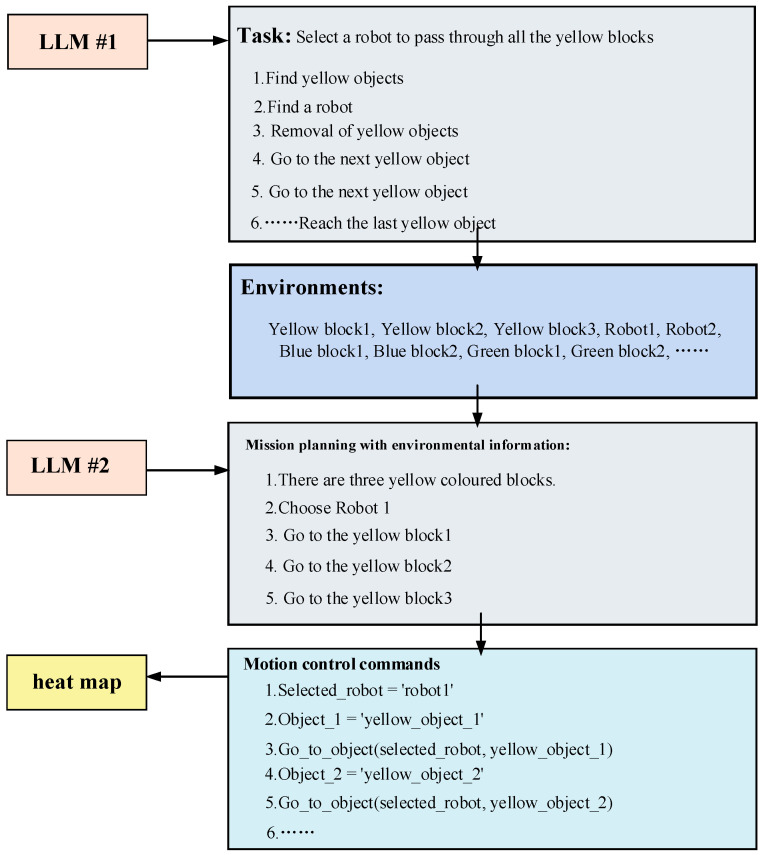
Example of the intermediate process of task planning.

**Figure 8 sensors-24-01687-f008:**
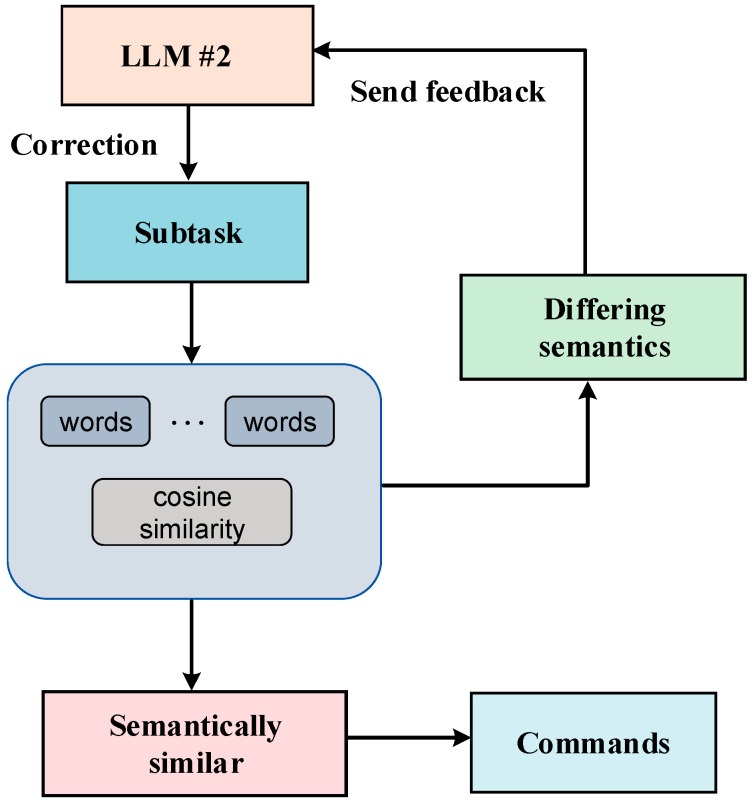
Idea of the cyclic semantic alignment method.

**Figure 9 sensors-24-01687-f009:**
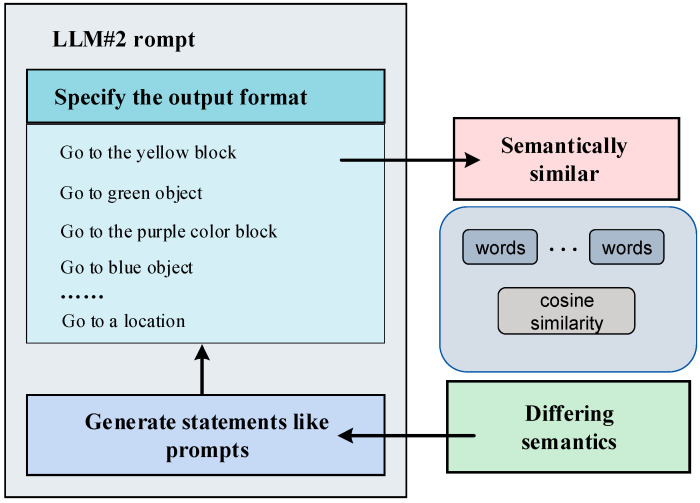
Example of a hint for LLM#2 in a cyclic semantic alignment approach.

**Figure 10 sensors-24-01687-f010:**
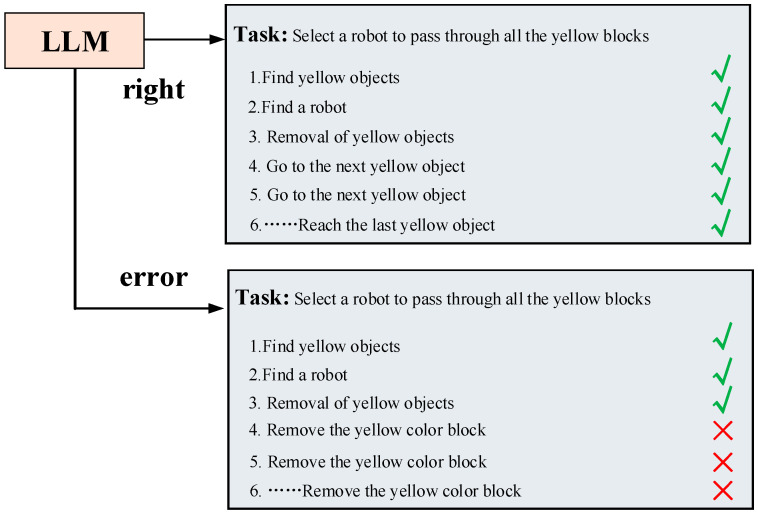
Example of LLM-generated policy error.

**Figure 11 sensors-24-01687-f011:**
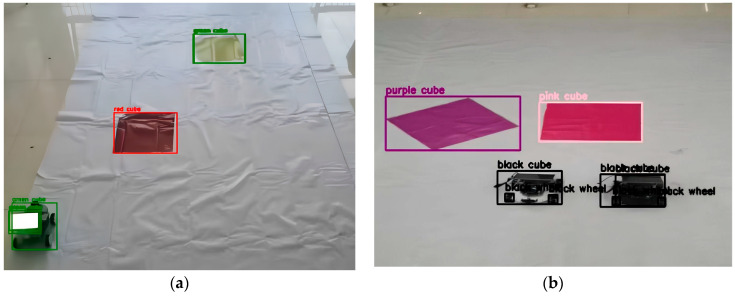
(**a**) VLM identifies both yellow-colored blocks and black-colored cars as green objects when there are changes in light brightness. (**b**) VLM will recognize the objects along with the attached blocks, such as identifying black wheel.

**Table 1 sensors-24-01687-t001:** In the middle of the two steps of task decomposition and vector alignment, different numbers of feedback attempts were set, and five tasks were performed to compare mandated execution success rates.

Task	Feedback 0	Feedback 1	Feedback 2	Feedback 3	Feedback 4	Feedback 5
Task 1: Travel to two target sites	4/10	7/10	8/10	9/10	10/10	10/10
Task 2: Travel to three target sites	3/10	6/10	8/10	8/10	8/10	8/10
Task 3: Planning the shortest route	3/10	6/10	7/10	8/10	8/10	8/10
Task 4: Self-determination of target location	4/10	5/10	7/10	7/10	7/10	7/10
Task 5: Multi-robot to multi-objective tasks	2/10	5/10	6/10	7/10	7/10	7/10
total	32%	58%	72%	78%	80%	80%

## Data Availability

Data are contained within the article.

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
