# Peer review of "Enhancing Robot Task Planning and Execution through Multi-Layer Large Language Models"

_sensors, 2024, doi:10.3390/s24051687_

Round 1

Reviewer 1 Report

Comments and Suggestions for Authors

The manuscript aims to develop a framework integrating the capabilities of LLMs into robot control tasks so that humans can interact with robots more naturally. To this end, a multi-stage process is proposed to use an LLM to decompose the human instruction/task query into coarse sub-tasks first, a second LLM is then used to decompose the sub-task into fine-grained robot actions. The idea is very interesting and can be very useful in many practical use cases. However, the experiment section lacks much essential information. Details of comments/suggestions are as follows.

--Line 266, how can normalising the text vectors remove the effect of text length? Please explain and clarify such an operation represented by Equation (2).

--It is unclear how the robot controls are defined. Please list all the defined robot action units used in the experiments.

--The details of LLMs used in the experiments are missing. How can different LLMs affect the results?

--The details of VLMs used in the experiments are missing. How can different VLMs affect the results?

--The time and memory complexity should be given and analysed together with the specs of the server/computer. What is the latency during the process of large models?

--More details of the experimental design, especially the hardware information, are needed. Please also show the real photos of the devices and experimental settings.

--It is strongly suggested to showcase some successful and failed cases for different tasks in the result section. Showcase the failure cases caused by different modules, e.g., the LLM, the VLM and semantic similarity modules, and present photos for each failure case.

--It will be more than useful to use a video clip to demonstrate how the experiments were conducted.

--There exist many language issues (i.e. inconsistency of terms) due to either using translation or language generation services. Language issues and typos: line 131, "Tellex et al. [9] Their work..." and similar citation issues throughout the manuscript; line 164, "macro-language model..."; line 187 "se-mantic similarity..."; line 190 "in-formation..."; line 260, "big language model..."; line 281, "thermal map..."; line 291, remove the last stop; lines 309,321 "calorific value map..."; lines 312, 319 "calorific values..."; line 387, "as shown in Fig ??"; 

Comments on the Quality of English Language

The manuscript needs thorough proofreading.

Author Response

Thank you very much for your affirmation and recognition of our work. We sincerely thank the editor and all reviewers for their valuable feedback that we have used to improve the quality of our manuscript. We have attached the revised detailed report for your reference.

Reviewer 2 Report

Comments and Suggestions for Authors

The focus of the paper is on enhancing robot task planning and execution through the use of multi-layer large language models (LLMs). It addresses the problem of limitations in current research in addressing complex tasks and proposes a multi-layer task decomposition architecture using LLMs to mitigate complexity and execution difficulty. . The architecture involves breaking down complex tasks into low-complexity tasks and integrating environmental information through a visual language model. The LLMs generate fine-grained tasks that align with robot control instructions at the semantic level, facilitating effective control of robots in tasks of heightened complexity. The document also introduces the heat value map navigation algorithm, which utilizes visual and language models to sense environmental information and guide robot behaviors.

 There are several ways mentioned to improve the paper:

  1. Enhancing the accuracy of task decomposition: The proposed multi-layer task decomposition architecture using Large Language Models (LLMs) can be further developed to provide even more precise guidance for robot behaviors through natural language.
  2. Incorporating more environmental perception: By adding a multi-environmental sensing module, the robot can obtain richer and three-dimensional environmental information through multi-modal environmental sensing, leading to improved task planning and execution.
  3. Optimization of motion control strategy: Constructing more complex numerical spaces can optimize the motion control strategy, enabling the robot to perform more delicate tasks.
  4. Optimization of the task decomposition process: By designing prompt words with more semantic information and utilizing prompt engineering, it is possible to reduce the number of cycles in the task decomposition process and optimize the overall model system.

Comments on the Quality of English Language

The paper's english should be enhanced. 

Author Response

(The authors gave the same response as above.)

Reviewer 3 Report

Comments and Suggestions for Authors

1.The article proposes a semantic alignment method. The proposed method is of great significance to the combination of the output of the large language model and the control of the robot. However, this method does not appear in the keywords. It is recommended to modify it;

2.Figure 1 provides an overview of the method proposed in the article, but it is recommended to further improve it. For example: the text format in each box needs to be unified. Currently, some first letters are capitalized and some are lowercase; the two "Breakdown of tasks" are different and need to be represented in the diagram;

3.In the Introduction, it is recommended to provide a concise expression of the contribution made in the article;

4.It is recommended that the idea of the cyclic semantic alignment method proposed in the article be introduced in the form of a diagram to facilitate readers to quickly understand;

5.Although the article mentions the construction of a real experimental verification platform, there are no relevant experiments on the real platform in the subsequent content. It is recommended to explain;

6.It is recommended to explain what the different colored squares in the heat value map represent to avoid confusion for readers;

7.It is recommended that the large language model and visual language model used in this article be briefly introduced in the experimental chapter;

8.In section 4.3, the author uses both "Feedback" and "Cycle", and it is recommended to unify;

9.In the paragraph below Figure 4, the sentence "as shown in Fig." needs to be improved.

Author Response

(The authors gave the same response as above.)

Round 2

Reviewer 1 Report

Comments and Suggestions for Authors

The authors have addressed all my concerns and the paper is ready to be accepted.

Author Response

Dear Reviewer,

I wanted to express my sincere gratitude for taking the time to review our paper. Your insightful comments and suggestions have significantly contributed to the improvement of our work. We have carefully considered and implemented your feedback, and we believe it will enhance the quality of the paper. Once again, thank you for your invaluable assistance and support.

Reviewer 3 Report

Comments and Suggestions for Authors

The author has done a lot of work, but there are still some questions:

1.           Visual Aid Integration:

·    Original: "Figure 5. Experimental design that describes the process of generating calorimetric map motion trajectories from image information."

·    Comment: Figures 5, 6, and 7 are referred to in the text. Consider incorporating brief descriptions or explanations of these figures in the main text to guide the reader.

2.           Task Complexity Description:

·    Original: "We designed five task scenarios, as shown in Fig. 6, in which we used different colored blocks to represent objects in the environment."

·    Comment: Provide a concise summary or overview of the five task scenarios at the beginning of this subsection to give the reader a clear understanding before delving into details.

3.           Real-world Application Implications:

· Comment: Provide a brief discussion or implication of how the findings from the experiments contribute to the real-world application and address challenges in natural language-based robotics.

Comments on the Quality of English Language

Ensure that the language used is clear and concise. Break down complex sentences for improved readability, especially when discussing technical details.

Author Response

I would like to express my heartfelt gratitude to you for taking the time to review our paper. Your insightful comments and suggestions have made a significant contribution to our work. We have carefully considered and implemented your feedback, believing it will enhance the quality of the paper. We have included the revised report in the appendix. Once again, thank you for your valuable input.
